# Design of a Femtosecond Laser Percussion Drilling Process for Ni-Based Superalloys Based on Machine Learning and the Genetic Algorithm

**DOI:** 10.3390/mi14112110

**Published:** 2023-11-17

**Authors:** Zhixi Zhao, Yunhe Yu, Ruijia Sun, Wanrong Zhao, Hao Guo, Zhen Zhang, Chenchong Wang

**Affiliations:** 1State Key Laboratory of Rolling and Automation, Northeastern University, Shenyang 110819, China; 2310245@stu.neu.edu.cn (Z.Z.); zhangzhen@stu.neu.edu.cn (Z.Z.); wangchenchong@ral.neu.edu.cn (C.W.); 2Shagang School of Iron and Steel, Soochow University, Suzhou 215137, China; 3AECC Hunan Aviation Powerplant Research Institute, Zhuzhou 412002, China; 13574274748@139.com; 4Science and Technology on Advanced High-Temperature Structural Materials Laboratory, Beijing Institute of Aeronautical Materials, Beijing 100095, China; zwr15632800829@163.com (W.Z.); neuzzzz@163.com (H.G.)

**Keywords:** femtosecond laser, percussion drilling, process design, machine learning, genetic algorithm

## Abstract

Femtosecond laser drilling is extensively used to create film-cooling holes in aero-engine turbine blade processing. Investigating and exploring the impact of laser processing parameters on achieving high-quality holes is crucial. The traditional trial-and-error approach, which relies on experiments, is time-consuming and has limited optimization capabilities for drilling holes. To address this issue, this paper proposes a process design method using machine learning and a genetic algorithm. A dataset of percussion drilling using a femtosecond laser was primarily established to train the models. An optimal method for building a prediction model was determined by comparing and analyzing different machine learning algorithms. Subsequently, the Gaussian support vector regression model and genetic algorithm were combined to optimize the taper and material removal rate within and outside the original data ranges. Ultimately, comprehensive optimization of drilling quality and efficiency was achieved relative to the original data. The proposed framework in this study offers a highly efficient and cost-effective solution for optimizing the femtosecond laser percussion drilling process.

## 1. Introduction

With the constant improvement in engine efficiency and increase in the thrust–weight ratio in the aerospace manufacturing field, turbine blade inlet temperatures are rising, resulting in higher service temperature demands [1,2]. However, even the most advanced nickel-based single-crystal superalloys cannot withstand these temperature requirements, making cooling solutions essential [3,4]. In this regard, film-cooling hole technology is widely used and can be considered an effective approach for ensuring higher turbine blade service temperatures [5,6].

Conventional methods for film hole processing include the use of long-pulse lasers, electric sparks and electrolyte beams [7,8]; however, femtosecond laser drilling has become the mainstream method for processing high-quality film-cooling holes [9]. Compared to traditional processing drilling methods, femtosecond laser drilling offers several advantages, such as fewer thermal defects, higher drilling precision and efficiency due to its extremely short pulse width (hundreds of femtoseconds) [10] and more concentrated laser energy [11]. However, the processing quality of film-cooling holes also significantly affects the service life of turbine blades to a large extent [12]. Achieving optimal processing parameters and ensuring the quality of micro-holes for femtosecond laser percussion drilling is challenging because of the complex nature of the coupled parameters involved [13].

Traditionally, the search for an optimized process is based on a trial-and-error approach, in which the process parameters and micro-hole quality are regularly explored. For example, A. Corcoran et al. [14] qualitatively investigated the effects of laser pulse energy and pulse width on the quality of femtosecond laser drilling holes through orthogonal experiments. They found that higher pulse energy and smaller pulse width reduced the generation of micro-cracks, thus improving hole quality. Nevertheless, the trial-and-error method involves numerous complex variables related to both thermal and non-thermal parameters, making the exploration time-consuming and costly [15,16]. In addition, globally extending the entire parameter space is challenging, resulting in limited optimization. In consequence, there is an urgent need to develop an efficient and cost-effective process optimization method to quickly determine the laser processing parameters. In recent years, machine learning, with its powerful data analysis capabilities, has been used to build prediction models between process parameters and performance objectives. Combined with global optimization algorithms, machine learning can find optimal plans on the basis of objective function, thus achieving high-efficiency and low-cost optimization. This approach has been widely applied in the fields of laser cutting and milling. For example, Chaki et al. [17] established a regression prediction model for the laser cutting of aluminum alloy using an artificial neural network and performed multi-objective optimization using a particle swarm optimization algorithm. Their results showed that the kerf width and surface roughness were decreased by 36.75% and 14.94%, respectively, and the material removal rate was increased by 24% with good performance enhancement. Addona et al. [18] used a CO_2_ laser and an artificial neural network algorithm to model the relationship between six-dimensional inputs and two-dimensional outputs in the milling of permanent magnets. Their model provided accurate predictions with an average absolute percentage error greater than 87%. Hence, it is evident that process design incorporating machine learning and optimization algorithms has been successfully applied in the field of laser material processing.

In this study, we propose a comprehensive process optimization framework for femtosecond laser percussion drilling using machine learning and a genetic algorithm (GA). Based on machine learning, the Gaussian support vector regression (G-SVR) model enables the accurate prediction of the relationship between process parameters and micro-hole quality. The GA is successfully applied to collaboratively and quantitatively design multi-parameters within and outside the original data ranges, achieving the comprehensive optimization of the taper and material removal rate (MRR) of micro-holes. This method provides a reliable and effective plan for the optimization of femtosecond laser percussion drilling.

## 2. Materials and Methods

### 2.1. Materials and Experimental Facilities

In this research, the DD6 nickel-based single-crystal superalloy was selected for its good performance and low cost, making it widely used for turbine blades. For the experimental setup, a commercial laser from Light Conversion in Lithuania, with a wavelength of 1030 nm, was applied for femtosecond laser percussion drilling. The laser spot diameter was set to approximately 55 μm. The intensity of the laser beam was distributed in a Gaussian space, and a beam splitter, along with a half-wave plate, was used to control the laser energy and monitor the laser power in real-time using a pyroelectric detector. The experimental equipment of drilling is shown in Figure 1.

A 0.6 mm thick plate made of DD6 nickel-based superalloy was used in the study. Before drilling, all sample surfaces were subjected to ultrasonic cleaning using ethyl alcohol to remove impurities. Percussion drilling was performed by focusing the laser beam on the surface of the samples. The entrance and exit diameters of the micro-holes induced by the laser drilling were observed and measured accurately using a scanning electron microscope. According to Equation (1), the taper of micro-holes can be acquired [19,20]:(1)$$Taper=tan^{-1}\left|\frac{\left(d_{entrance}-d_{exit}\right)}{2h}\right|$$
where $d_{entrance}$ and $d_{exit}$ represent the entrance and exit diameters, respectively; h is the depth of the micro-holes.
(2)$$MRR=\Delta m/\left(\rho \cdot n\right)$$

Considering the difficulty of measuring the quality of individual micro-holes due to their small size, we calculated the total sum of the quality changes and their average across 60 micro-holes [21,22]. Subsequently, we recorded the number of pulses and calculated the volume changes using the material density to determine the MRR of each individual micro-hole. According to Equation (2), the MRR of micro-holes can be acquired, where Δm represents the change in mass, ρ represents the density of the DD6 alloy and n represents the number of laser pulses, respectively.

### 2.2. Simulation of Machine Learning

In this paper, according to the scikit-learn platform in Python^®^, five kinds of machine learning algorithms were selected to construct the regression prediction models. These algorithms included support vector regression (SVR), multilayer perceptron (MLP), random forest (RF), gradient boosting regression (GBR) and extreme gradient boosting (XGB). The SVR algorithms were divided into linear and Gaussian support vector regression depending on the kernel function used in this research [23]. All selected algorithms are suitable for addressing the data problem addressed in this study.

It was necessary to standardize the dataset to eliminate any dimensional differences among the features. Furthermore, we needed to divide the dataset into a training set and a testing set using a ratio of 8:2 prior to regression establishment. By randomly partitioning the dataset 50 times, the average prediction results and standard deviation results were obtained. The optimal ratio and number of divisions were determined through a series of pre-tests. To objectively evaluate the generalization ability of the various models and select the optimal ones, the squared correlation coefficient (*R*^2^) and mean absolute error (MAE) of the prediction results were calculated, as follows [24,25]:(3)$$R^{2}=\frac{{\left(n\sum_{i=1}^{n}f\left(x_{i}\right)y_{i}-\sum_{i=1}^{n}f\left(x_{i}\right)\sum_{i=1}^{n}y_{i}\right)}^{2}}{\left(n\sum_{i=1}^{n}f{\left(x_{i}\right)}^{2}-{\left(\sum_{i=1}^{n}f\left(x_{i}\right)\right)}^{2}\right)\cdot \left(n\sum_{i=1}^{n}y_{i}{}^{2}-{\left(\sum_{i=1}^{n}y_{i}\right)}^{2}\right)}$$
(4)$$MAE=\frac{1}{n}\sum_{i=1}^{n}\left|f\left(x_{i}\right)-y_{i}\right|$$
where n is the number of samples; $f\left(x_{i}\right)$ and $y_{i}$ express the predicted values and experimental values of the ith sample, respectively. Undoubtedly, the goal was to obtain an algorithm model with an *R*^2^ value close to 1 and with a low MAE value, indicating a higher level of accuracy in the results. The experimental procedure of this research is shown in Figure 2.

## 3. Results and Discussion

### 3.1. Dataset Collection

The above percussion drilling experiments were conducted to generate a dataset consisting of 81 sets of data. These data included four process parameters: laser power, pulse width, frequency and defocusing amount. In addition, two performance parameters, namely taper and MRR, were measured. The selection of the process parameter ranges was based on a series of pre-tests and equipment feasibility, with three different levels set for each parameter. Specifically, the laser power levels were 5, 10 and 20 W, the pulse width levels were 300, 500 and 800 fs, the frequency ranged from 100 to 200 kHz in increments of 50 kHz, and the defocusing amount ranged from −150 to 150 μm in increments of 150 μm. Also, taper and the MRR were used to characterize hole quality and efficiency, respectively. In this regard, this study aimed to minimize taper while maximizing the MRR. To investigate the effects of the various process parameters on taper and the MRR, a full factorial design (3^4^ = 81) was employed. Table 1 presents the critical statistics for the model features derived from the experimental data.

### 3.2. Selection of Machine Learning Algorithms

This study implemented generalization ability comparison and overfitting analysis to select the machine learning methods with good performance. Overfitting refers to making too stringent assumptions, resulting in excellent predictions in the training set but poor results in the testing set. In contrast, underfitting occurs when the model fails to fit the data well, resulting in poor results in both the training and testing sets [26,27].

The four process features including power, pulse width, frequency and defocusing amount were used as model inputs, while taper and the MRR were treated as predictors of the model. Fifty models were built using an 8:2 division of the data into training and testing sets, and 4-fold cross-validation was employed to ensure the validity and robustness of the results. The *R*^2^ and MAE values of the models were compared in order to analyze the prediction ability of the different algorithms, as portrayed in Figure 3. Each bar graph represents the average of 50 divisions, with error bars indicating the general nature of the models and an absence of outstanding individual results. From the *R*^2^ and MAE figures, it can be observed that L-SVR has the poorest prediction accuracy for both taper and MRR, indicating its inability to effectively address the problems identified in this study. MLP and RF exhibit severe overfitting phenomena in both the training and testing sets. In addition, MLP shows a long error bar, suggesting high instability. In contrast, the G-SVR, GBR and XGB methods demonstrate excellent performance, indicating no significant overfitting or underfitting issues. The *R*^2^ of the taper values is 80% and 60% in the training and testing set, while the MAE is 0.7° and 1.1°); for the MRR, the *R*^2^ is 90% and 85% in the training and testing set, with a MAE of 85 and 125 μm^3^·pulse^−1^, respectively. It is worth mentioning that there is no significant difference in the predictions of the MRR among these three algorithms, although G-SVR is slightly less accurate and stable than GBR and XGB when predicting taper.

To further assess the extent of overfitting in each algorithmic model, an analysis was conducted on the 50 model prediction results mentioned earlier. This analysis was based on the absolute difference between the *R*^2^ values of the training and testing sets. The results were classified into three levels: 10~20%, 20~30% and >30%. Figure 4 illustrates the number of overfitting models at each level, with larger values indicating more severe overfitting problems. In particular, MLP and RF produced over 35 overfitting models in the 20~30% and >30% ranges for taper, indicating significant overfitting issues. Although the number of overfitting models for the MRR was not as high in the 20~30% and >30% ranges, MLP and RF still produced over 30 overfitting models across all three ranges (>10%), compared to less than 20 for the other four algorithms. This suggests an extremely severe overfitting problem, which is consistent with the previous analysis. In contrast, the remaining four algorithms, namely L-SVR, G-SVR, GBR and XGB, produced relatively low numbers of overfitting models. This indicates their potential for accurate prediction and further optimization, except for L-SVR, which exhibited very low accuracy. Thus, G-SVR, GBR and XGB were selected to further optimize the subsequent prediction model.

### 3.3. Feature Analysis and Building Prediction Models

To further improve the accuracy of the models and reduce overfitting, this study used a dimensionality reduction method, as outlined in a previous study [28]. The dimensionality reduction process in this study involved removing features with low correlation between inputs and outputs based on Pearson correlation coefficient and mean decrease accuracy (MDA) analyses. The Pearson correlation coefficient is defined as the covariance between two variables divided by the product of their standard deviations. This measures the degree of linear correlation between different variables, ranging from −1 to 1, with absolute values closer to 1 indicating a stronger correlation [29]. MDA analysis involves disrupting the order of input parameters and observing the effect on output characteristics. Larger values indicate a greater influence on performance parameters [30].

The results of the Pearson correlation coefficient and MDA analyses are illustrated in Figure 5. All of the results demonstrate tangible values, confirming the rationality of the selected parameters. Depending on the shade of the color and the magnitude of the values, the influence of the input parameters on the relevance and importance of the output parameters can be determined. Among the four process parameters, frequency has the lowest value for both taper and the MRR; therefore, it is apparent that the frequency feature has the least relevance and importance for model performance.

According to the results of the feature analysis, only the three-dimensional features of power, pulse width and amount of defocus were selected as inputs, excluding the feature of frequency. Three algorithms, including G-SVR, GBR and XGB, were used to establish prediction models based on the findings described in Section 3.2. The dataset was randomly divided using an 8:2 ratio of training set to testing set, and the models were trained 50 times. Figure 6 illustrates the *R*^2^ prediction results with MAE values for the three algorithms (G-SVR, GBR and XGB) for both taper and the MRR. For taper, all three algorithms showed that the mean values of *R*^2^ and MAE were concentrated at 70% and 0.87° in the training set and 62% and 1.05° in the testing set, respectively. As for the MRR, it is evident that the models exhibited higher accuracy and greater stability, achieving 90% and 100 μm^3^·pulse^−1^, and 85% and 120 μm^3^·pulse^−1^, in the training and testing sets, respectively. However, the G-SVR algorithm demonstrated the lowest MAE value for the MRR and the least overfitting for taper. Therefore, the G-SVR model was chosen as the optimal algorithm for the final prediction of taper and the MRR. This was then combined with the GA for the subsequent process optimization.

### 3.4. Process Design within the Range of the Original Dataset

The specific GA procedure used in this study is portrayed in Figure 7, providing an approximate explanation of the basic principles involved, which consist of four processes [31]. First, a population of a certain size is randomly generated including different individuals within the specified ranges. Each individual in the population represents a combination of laser power, pulse width and defocusing amount. Second, the objective (taper or the MRR) is calculated using the G-SVR prediction models and sorted based on the fitness function value. Third, genetic manipulation is applied, including selection, crossover and mutation inspired by biological principles, to improve the population by favoring individuals with better objective function values. An elite strategy is introduced to preserve individuals with the highest fitness values in each generation, enhancing the quality of the population. Simultaneously, crossover and mutation introduce new individuals and explore unexplored regions of the solution space. As the number of genetic generations increases, the results of each generation become progressively superior to the previous one until convergence. Finally, the termination condition is met when the results stabilize after several genetic generations.

Since this experiment addresses taper and MRR performance, constituting a multi-objective problem, the NSGA-II genetic algorithm (non-dominated sorting GA of the second generation) [32] was proposed. This algorithm follows a similar principle to single-objective optimization, with the only difference being in the calculation and sorting of the objectives in step 2. For single-objective optimization, the objectives are chosen directly based on their values, while for multiple-objective optimization, the objectives are selected based on the Pareto front and congestion option. Furthermore, the hyperparameters for the GA used in this study—for both single-objective and multi-objective outputs—were set as listed in Table 2. The optimal configurations of these hyperparameters were determined through an exhaustive method that considered the population size, number of generations, crossover and mutation. This ensured the efficient and accurate attainment of the optimal solution during the evolutionary process for subsequent performance optimization.

Based on the results presented in Section 3.3, G-SVR was deemed most suitable for further optimization and design in conjunction with the GA. Using *R*^2^ as the criterion, the G-SVR regression prediction models were filtered out from the initial set of 50 models; only models with high testing-set *R*^2^ values and no overfitting for both taper and the MRR were retained, whereas illogical and less accurate results were discarded. Subsequently, single-objective performance optimization was conducted for both taper and the MRR. Figure 8a,b depicts the results of this optimization compared with the original data. All the taper results were, evidently, smaller than the original values, concentrated around 3°. Furthermore, the MRR consistently achieved excellent results, with values predominantly around 1800 μm^3^·pulse^−1^.

To further explore the synergistic optimization of taper and the MRR, a high throughput multiple-objective GA was employed, enabling quality and efficiency in percussion drilling. Figure 8c,d illustrates the outcomes of the comprehensive optimization, represented by the blue dots, as well as the optimal design result for the process and performance. Comparing these results to the best values obtained from the original dataset for both taper and the MRR, it is evident that the power remains unchanged, while the pulse width and defocusing amount increase by 167% and 125 μm, respectively. Therefore, the optimization process generated innovative process parameters that are notably distinct from the original data. Furthermore, substantial improvements were observed for both taper and the MRR, with a decrease of 68% and an increase of 42%, respectively.

### 3.5. Process Design Outside of the Range of the Original Dataset

According to the optimization results within the original data range presented in Section 3.4, impressive optimization and design were achieved considering both taper and the MRR. To further enhance the generalizability of the design in this study, a process design beyond the original data range is proposed, with the selection scope based on the feasibility of the devices.

Starting with the single-objective performance optimization for taper and the MRR separately, the results demonstrate a higher degree of optimization. The taper optimization yielded a maximum value of 2.8° and a minimum value of just 0.2°, which is even lower than the minimum achieved within the original data range, let alone that compared with the original data. Regarding the MRR, the externally optimized results fall within the range of 1800~2400 μm^3^·pulse^−1^, surpassing the previous internal optimization range of 1700~1800 μm^3^·pulse^−1^. This significantly improves and exceeds the maximum MRR of 1888 μm^3^·pulse^−1^ observed in the original data. Consequently, these outcomes undeniably demonstrate that optimization beyond the original data range can produce higher-quality film-cooling holes.

The NSGA-II algorithm was used for the drilling design beyond the original data range, and the optimization results are depicted in Figure 9c,d. It can be clearly seen that within this extended range, the taper of the holes can be reduced to 0~1°, while the MRR can be increased to 2500~2700 μm^3^·pulse^−1^. This clearly indicates the formation of a favorable Pareto front, representing an exceptionable result. Subsequently, the optimal dataset was selected from several Pareto results, in the upper right corner. In this case, the power is 36 W, the pulse width is 1000 fs and the defocusing amount is 184 μm, while the taper is reduced by 93% and the MRR is increased by 107% compared to the original data optimum. Furthermore, compared to the optimization results within the original data range, the taper is reduced by 79% and the MRR is increased by 46%. This clearly demonstrates the feasibility of applying this process solution in practical percussion drilling. In other words, the process of femtosecond laser percussion drilling can be optimized and designed efficiently and cost-effectively, facilitating the collaborative optimization of taper and the MRR.

In order to verify the reliability of the optimized process, experimental validation was performed outside the range of the original dataset. Using the process parameters designed outside the range of the original dataset, a SEM image of the resulting micro-hole was obtained, as shown in Figure 10a. The taper of this hole is 0.7° and the MRR is 2515 μm^3^·pulse^−1^. In contrast, using the original dataset with 81 sets of data, the optimum hole drilled by the femtosecond laser was selected and a corresponding SEM image was obtained. The taper of this hole is 6.8° and the MRR is 1280 μm^3^·pulse^−1^, as shown in Figure 10b. Compared with the original micro-hole, the optimized hole was greatly improved both in terms of taper and the MRR; taper was reduced by 90% and the MRR was increased by 96%, which is almost consistent with the optimization results outside the range of the original dataset. Therefore, these experimental results validate the reasonability of the design process and achieve the comprehensive optimization of quality and efficiency.

## 4. Conclusions

In this study, a multi-parameter collaborative optimization design framework for femtosecond laser percussion drilling was successfully established. Additionally, optimization was carried out within and outside the range of the original dataset, resulting in comprehensive improvements in the taper and MRR of micro-holes. This approach provides an efficient and low-cost performance optimization and process design. The primary conclusions are as follows:Through a comparison of generalization ability and overfitting analysis, six machine learning methods were gradually refined. G-SVR, GBR and XGB were selected for the further establishment of the prediction model owing to their good levels of accuracy and stability. In addition, it was observed that the frequency feature had the lowest Pearson correlation coefficient and MDA value for both taper and the MRR.To enhance the model accuracy and reduce overfitting, G-SVR, GBR and XGB prediction models were established using three input features, excluding the frequency feature based on the feature analysis. Among these, the G-SVR prediction model demonstrated higher accuracy for taper and the MRR; the *R*^2^ values using the training set and testing set were 69.5% and 62.9% for taper, and 90% and 85.8% for the MRR, respectively, while the MAE values in the training set and testing set were 0.865° and 1.03° for taper, and 97.1 and 115 μm^3^·pulse^−1^ for MRR, respectively.Combined with the GA, the double-objective performance was optimized within the range of original data. The optimal result for taper was reduced by 68%, while the MRR was increased by 42% compared to the original optimal data.To achieve more generalized design outcomes, the GA was employed to optimize the double-objective performance outside the original data range. As a result, taper and the MRR were reduced and increased by 79% and 46% respectively, compared to the optimal results within the original data range. Furthermore, taper and the MRR were reduced and increased by 93% and 107%, respectively, compared to the original optimum data.

## Figures and Tables

**Figure 1 micromachines-14-02110-f001:**
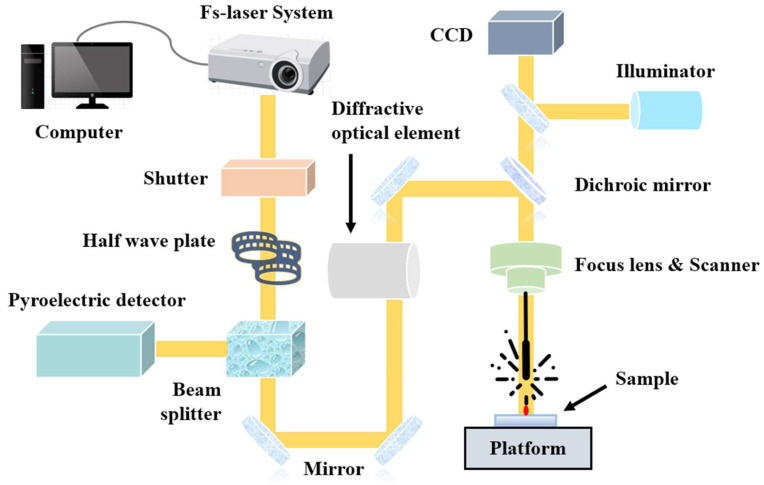
Femtosecond laser percussion drilling experimental equipment.

**Figure 2 micromachines-14-02110-f002:**
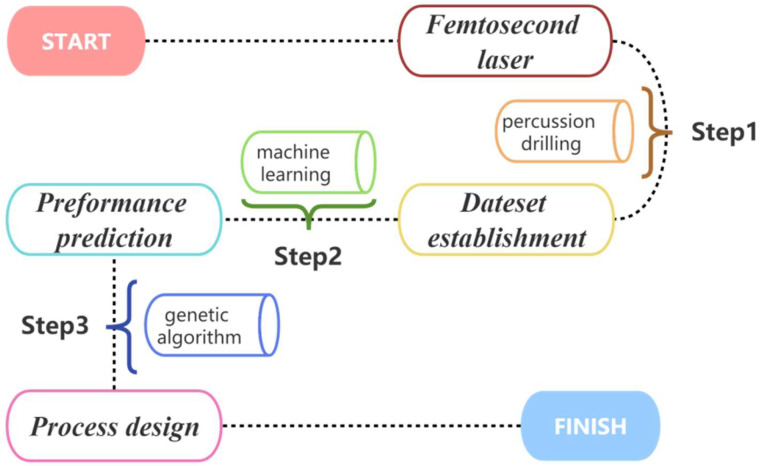
Experimental procedure of this study.

**Figure 3 micromachines-14-02110-f003:**
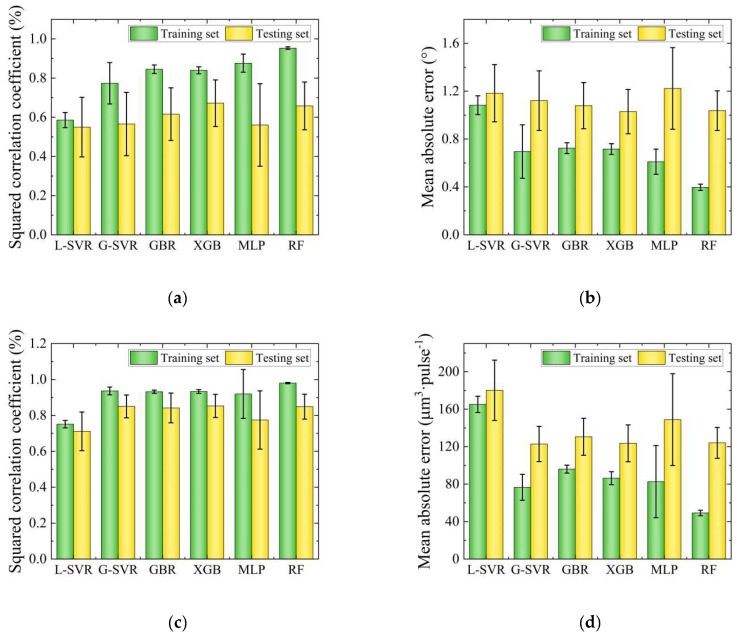
Results of different models using the initial six methods: (**a**) Results of R^2^ for taper; (**b**) results of MAE for taper; (**c**) results of *R*^2^ for the MRR; (**d**) results of MAE for the MRR.

**Figure 4 micromachines-14-02110-f004:**
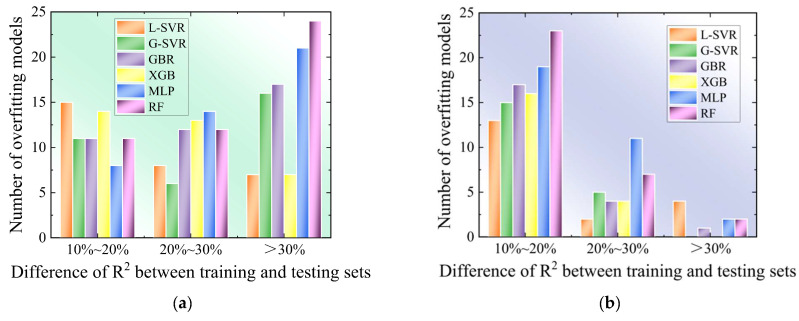
Number of overfitting models among various algorithms: (**a**) Outcomes for taper; (**b**) outcomes for the MRR.

**Figure 5 micromachines-14-02110-f005:**
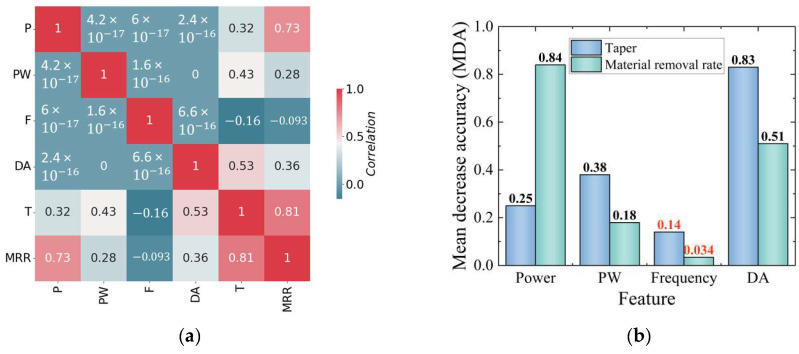
Feature analysis between process parameters and performance parameters: (**a**) Pearson correlation coefficient; (**b**) MDA.

**Figure 6 micromachines-14-02110-f006:**
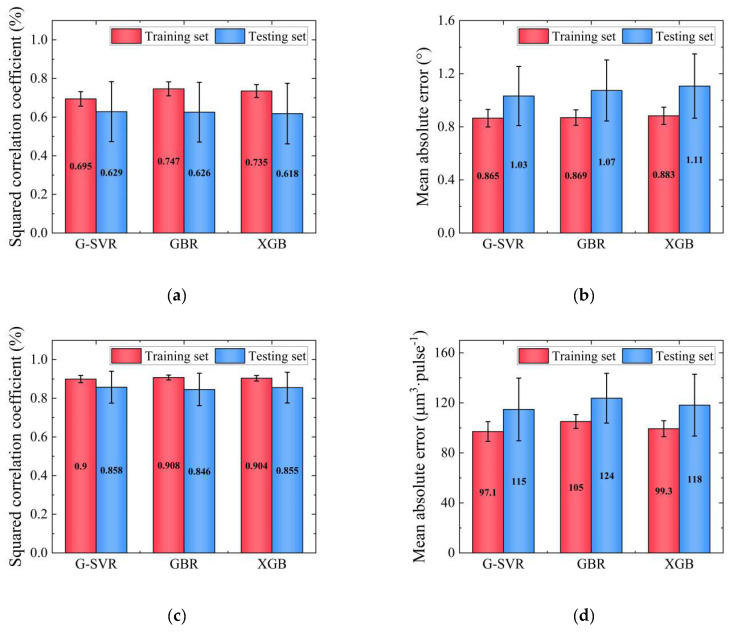
Mean prediction results for G-SVR, GBR and XGB models using machine learning: (**a**) Results of *R*^2^ for taper; (**b**) results of MAE for taper; (**c**) results of *R*^2^ for the MRR; (**d**) results of MAE for the MRR.

**Figure 7 micromachines-14-02110-f007:**
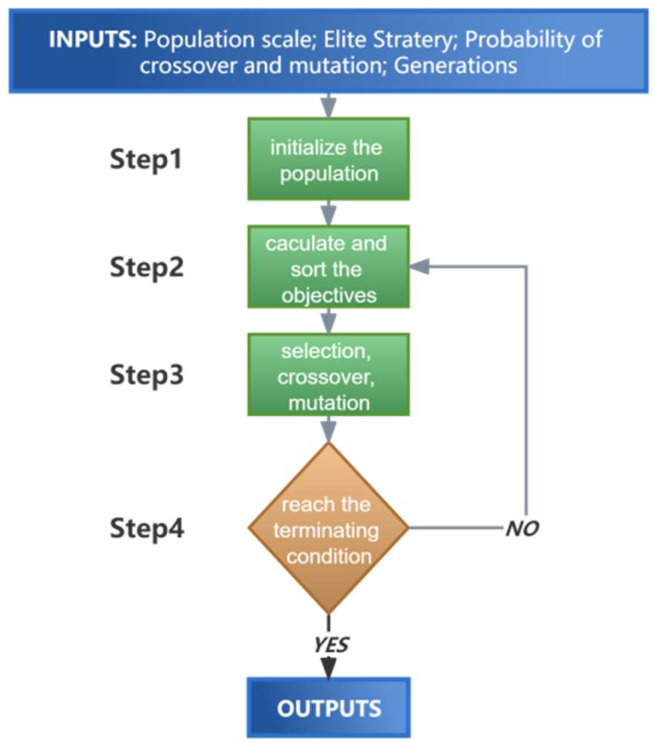
Flow chart of the GA.

**Figure 8 micromachines-14-02110-f008:**
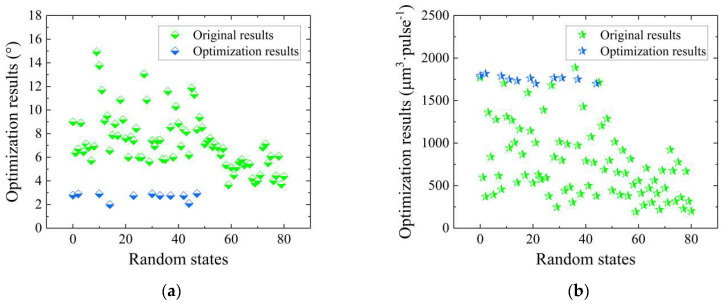
Optimization and design results within the range of the original dataset: (**a**) Optimization results for taper; (**b**) optimization results for the MRR; (**c**) optimization results for both taper and the MRR; (**d**) design results compared to the original results.

**Figure 9 micromachines-14-02110-f009:**
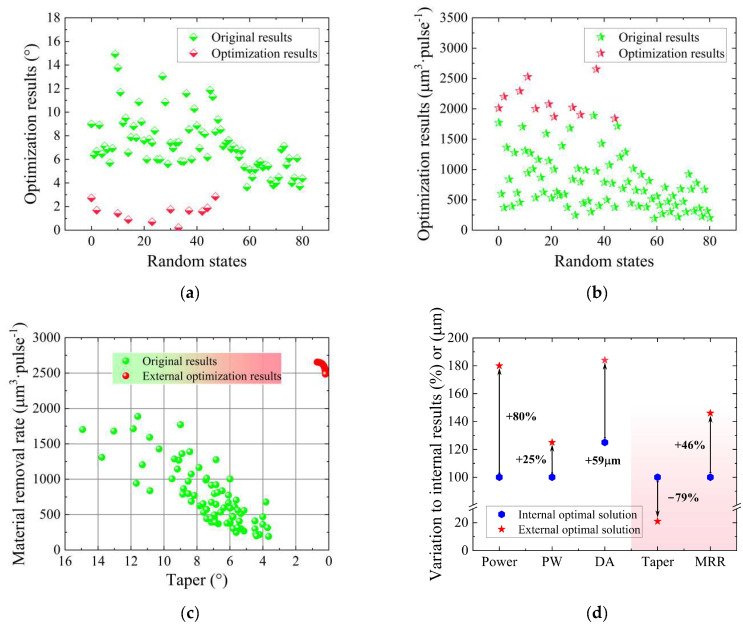
Optimization and design results outside the range of the original dataset: (**a**) Optimization results for taper; (**b**) optimization results for the MRR; (**c**) optimization results for both taper and the MRR; (**d**) design results compared to the original results.

**Figure 10 micromachines-14-02110-f010:**
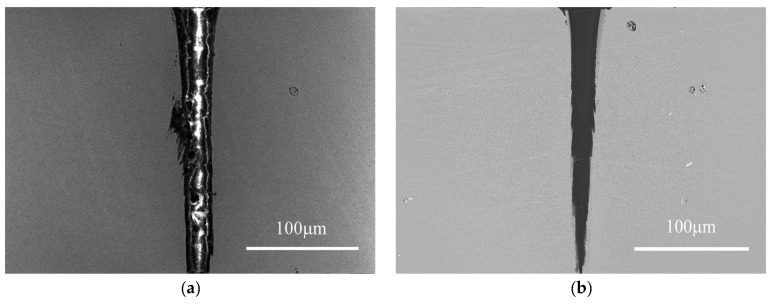
SEM images of femtosecond laser drilling: (**a**) Micro-hole with optimization; (**b**) micro-hole without optimization.

**Table 1 micromachines-14-02110-t001:** Inputs and outputs from experimental data.

	Variables	Maximum	Minimum	Average	Standard Deviation
Inputs	Power (W)	20	5	11.67	6.28
	Pulse width (fs)	800	300	533.33	206.76
	Frequency (kHz)	200	100	150	41.08
	Defocusing amount (μm)	150	−150	0	123.24
Outputs	Taper (°)	14.93	3.65	7.28	2.34
	Material removal rate (μm^3^·pulse^−1^)	1887.67	192.21	757.68	421.37

**Table 2 micromachines-14-02110-t002:** GA hyper-parameter optimization results.

Type	Population	Generation	Crossover	Mutation
Single-objective optimization	50	200	0.9	0.09
Double-objective optimization	100	500	0.9	0.1

## Data Availability

The data presented in this paper are available on request from the corresponding author.

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
