# Peer review of "Design of a Femtosecond Laser Percussion Drilling Process for Ni-Based Superalloys Based on Machine Learning and the Genetic Algorithm"

_micromachines, 2023, doi:10.3390/mi14112110_

Round 1

Reviewer 1 Report

Comments and Suggestions for Authors

In this manuscript, the authors put forward a systematic design process to optimize the laser percussion drilling parameters by combining machine learning and genetic algorithm. 6 machine learning algorithm were first compared to determine the optimal prediction model. Then comprehensive optimization of taper and material removal rate is conducted within and without the original data by employing genetic algorithm. Although I agree with the authors that the strategy can lead to efficient performance optimization compared with the traditional trial-and-error practice, this work lacks experimental validation to confirm this. My suggestions are as follows:

1.  The authors establish models to compare the prediction ability among different algorithms before performing feature analysis to remove features with low relevance. Why not first perform the feature analysis to exclude the influence of frequency parameter and then establish models to evaluate the performance of different algorithms, thus making the evaluation more accurate? Also, the negligible influence of frequency on fabrication results may be confirmed by first manual analysis of the dataset.

2. The calculation of material removal rate should be explained in more detail. It is recommended to provide the typical SEM image of the as-prepared micro-holes to add the readers’ visual understanding of quality measurement.

3. Are the substantial improvements in taper and material removal rate achieved by genetic algorithm only obtained in theory? Can the authors provide comparison of real fabrication results with and without the algorithm optimization? Only with the photos of fabricated micro-holes can the authors illustrate the effectiveness of the optimal fabrication parameters and the method in this work.       

Reviewer 2 Report

Comments and Suggestions for Authors

Title: Design of Femtosecond Laser Percussion Drilling Process in Ni-based Superalloys Based on Machine Learning and Genetic Algorithm

The authors presented a comprehensive research on the machine learning for femtosecond laser percussion drilling processing. The genetic algorithm, including support boosting regression, multilayer perceptron, random forest, gradient boosting regression and extreme gradient boosting, were applied to optimize the performances. The manuscript was well written and the conclusions were supported by the experimental results. The conclusions may be interesting for the researcher in the field of laser machining. I suggest to include some (SEM) images of laser drilling on Ni-based superalloys, which may help for the understanding on this study.

Round 2

Reviewer 1 Report

Comments and Suggestions for Authors

Thank you for dealing with all my questions carefully. I have no further questions now. This manuscript can now be accepted for publication in Micromachines.